# Insulin management in hospitalized patients with diabetes mellitus on high-dose glucocorticoids: Management of steroid-exacerbated hyperglycemia

Yu-Chien Cheng[1,2], Yannis Guerra[1,2]*, Michael Morkos[1,2], Bettina Tahsin[1,2], Chioma Onyenwenyi[1,2], Louis Fogg[3], Leon Fogelfeld[1,2]

1 Division of Endocrinology, John H. Stroger. Jr. Hospital of Cook County, Chicago, Illinois, United States of America, 2 Section of Endocrinology, Rush University Medical Center, Chicago, Illinois, United States of America, 3 Department of Community Nursing, Rush University Medical Center, Chicago, Illinois, United States of America

* yguerra@cookcountyhhs.org

**Data Availability Statement:** Data can be found on the Open Science Framework at https://osf.io/dzhu7/ (DOI: 10.17605/OSF.IO/DZHU7).

## Abstract

### Background

Glucocorticoid (GC)-exacerbated hyperglycemia is prevalent in hospitalized patients with diabetes mellitus (DM) but evidence-based insulin guidelines in inpatient settings are lacking.

### Methods and findings

Retrospective cohort study with capillary blood glucose (CBG) readings and insulin use, dosed with 50% basal (glargine)-50% bolus (lispro) insulin, analyzed in hospitalized patients with insulin-treated DM given GC and matched controls without GC (n = 131 pairs). GC group (median daily prednisone-equivalent dose: 53.36 mg (IQR 30.00, 80.04)) had greatest CBG differences compared to controls at dinner (254±69 vs. 184±63 mg/dL, P<0.001) and bedtime (260±72 vs. 182±55 mg/dL, P<0.001). In GC group, dinner CBG was 30% higher than lunch (254±69 vs. 199±77 mg/dL, P<0.001) when similar lispro to controls given at lunch. Bedtime CBG not different from dinner when 20% more lispro given at dinner (0.12 units/kg (IQR 0.08, 0.17) vs. 0.10 units/kg (0.06, 0.14), P<0.01). Despite receiving more lispro, bedtime hypoglycemic events were lower in GC group (0.0% vs. 5.9%, P = 0.03).

### Conclusions

Since equal bolus doses inadequately treat large dinner and bedtime GC-exacerbated glycemic excursions, initiating higher bolus insulin at lunch and dinner with additional enhanced GC-specific insulin supplemental scale may be needed as initial insulin doses in setting of high-dose GC.

**Funding:** The author(s) received no specific funding for this work.

**Competing interests:** The authors have declared that no competing interests exist.

## Introduction

Hyperglycemia is associated with increased morbidity and mortality in non-critically ill hospitalized patients [1–3]. Similarly, glucocorticoid (GC)-exacerbated hyperglycemia is shown to increase morbidity, infection rates, and hospital stay [4]. Given the increasing prevalence of diabetes mellitus (DM) and widespread use of GC for various therapeutic effects in inpatient settings, establishing an appropriate protocol for treating GC-exacerbated hyperglycemia in patients with DM becomes imperative to prevent poor clinical outcomes.

Moreover, the recent results of the RECOVERY trial, that showed the effectiveness of high-dose dexamethasone in patients hospitalized with Covid-19 who required oxygen, have increased the frequency of use of this medication [5]. As an example, in our institution, the profuse use of GC for patients with Covid-19 has been causing destabilization of glucose control and the need for much higher doses of insulin for these patients. Initial data show that the percentage of patient days with average blood glucose greater than 300 mg/dl more than doubled.

GC exacerbate hyperglycemia through several mechanisms. GC stimulate hepatic glucose production and protein catabolism and impair insulin secretion and glucose uptake of the peripheral tissues. GC effects are most pronounced postprandially with skeletal muscle responsible for 80% of postprandial glucose uptake [6–8]. In small clinical studies, significant GC-exacerbated postprandial hyperglycemia was confirmed through serial blood glucose monitoring [9], continuous glucose monitoring [10, 11] and the observed higher prandial insulin requirement for normoglycemia attainment in hospitalized patients with DM on GC [12]. However, GC effects on fasting hyperglycemia are only mild [10, 13, 14].

Despite the importance for controlling GC-exacerbated hyperglycemia in hospitalized patients with DM [6], underscored by a recent call to action for clinical research on inpatient GC-exacerbated hyperglycemia management [15], there is scant high-quality evidence or consensus regarding the optimal inpatient insulin regimen [15, 16]. Current proposed evidence-based approaches include: standard basal-bolus therapy with dexamethasone use [6, 17], standard basal-bolus therapy with the addition of NPH at the initiation of GC [18, 19] or the addition of NPH at three fixed times per day with multiple daily short-acting insulin [18], the use of NPH instead of glargine as the basal therapy [20] with greater proportion of bolus insulin between midday and midnight with prednisolone use [16], and the tailoring of the type and timing of insulin used depending on the steroid prescribed [21]. Nevertheless, none of these regimens has been shown to be superior to the usual care in achieving better overall glucose control in the inpatient settings.

Not only is the optimal insulin protocol unclear but optimal dosage also remains elusive [22]. While the Endocrine Society guideline in 2012 [6] recommends insulin initiation at 0.3–0.5 units/kg/day with GC use, no specific recommendation regarding the breakdown or proportion of basal versus bolus insulin was mentioned. The 2021 American Diabetes Association Standards of Medical Care [23] also did not provide specific starting insulin dosages, although increasing prandial and supplemental insulin in addition to basal insulin in higher doses of GC was suggested. Finally, studies examining dosage requirements to achieve normoglycemia had small sample sizes, different baseline demographics and degrees of glucose control prior to hospitalizations [12, 18]; therefore, the optimal doses found in these studies are ungeneralizable.

Given the high frequency of GC use in hospitals by providers of all specialties, there is a strong need for a uniform, straightforward, efficacious insulin protocol for all types of GC, ideally based on the basal-bolus regimen most familiar to most providers. Therefore, we aimed to gain further insight into the insulin use pattern and glucose control in patients with DM

treated with GC in an inpatient setting. Our inpatient hyperglycemia insulin protocol recommends a well-standardized 50% basal (glargine)-50% bolus (lispro) therapy (see S1 File). In this study, we performed a retrospective review to compare the insulin requirement and capillary blood glucose (CBG) control between hospitalized patients with DM with GC use and those without. We hypothesized that higher prandial insulin but similar basal insulin was given with GC use in the hospital.

## Methods

### Study design

This was a retrospective cohort study conducted at a single U.S. institution. The Institutional Review Board of the John H. Stroger, Jr. Hospital of Cook County, Chicago, Illinois, approved the study protocol.

### Participants and procedures

Data was collected from two databases: 1) electronic data warehouse (Cerner PowerChart®) provided the eligible hospitalizations with demographic and clinical information including hemoglobin A1C (HbA1c) and glomerular filtration rate (eGFR) on admission, doses and types of insulin and GC with corresponding dates and times; 2) hospital glucose database collected for inpatient diabetes quality assurance provided CBG with corresponding times and dates. The collected information per hospitalization from both databases was combined for analysis.

Data was extracted from all inpatient stays to identify patients meeting the inclusion criteria of those with DM who were non-pregnant, aged ≥18 years and who received any insulin from 1/1/2015 to 12/31/2015. We used inpatient data from 2015 as a convenience sample as previous studies [13, 15, 23] have employed smaller samples.

Exclusions included hospitalizations of less than 3 days (3 days is the minimal needed for reasonable insulin titration); admittance to any intensive care unit (ICU) or eGFR<30 mL/min/1.73 m$^2$ on admission (we have separate hyperglycemia protocols for such patients); missing weight on admission to express insulin in units/kilogram (kg); hospitalized patients not treated per protocol; those with HbA1c <6.5% (47.5 mmol/mol) to include only patients with DM.

We divided eligible patients into two groups based on whether they received GC treatment during hospitalization (GC group vs. non-GC control group). We matched those in the GC group, using one-to-one basis, to those in the control by the following variables: age (±10 years), gender, race, and HbA1c (±0.5% given the inter-individual variability) [24].

### Measures

**Hospitalization days.** Data on CBG and insulin doses were represented by hospitalization day, with the admission day counted as hospitalization day 0, the next day as hospitalization day 1 and so on. We included CBG and insulin doses up to day 6 of the hospitalizations since 75% of our hospitalizations had a length of stay up to 7 days.

**Glucocorticoids use.** In the GC group, days without receiving GC treatment were eliminated. GC were converted into prednisone-equivalent doses in mg and mg/kg [25, 26], and GC dosage were totaled per hospitalization day. This was done to be able to compare patients across different therapies or to accomodate those who received different types of steroids in the same hospitalization. Due to hospitalization day being the unit of analysis, the conversion allowed us to compare the different doses of different steroids on various days.

Timing of GC administration was categorized into four times based on common time blocks for daily activities: 12:00 am to 5:59 am, 6:00 am to 11:59 am, 12:00 pm to 5:59 pm, and 6:00 pm to 11:59 pm.

**Capillary blood glucose.** All CBG readings obtained were used. Beyond the recommended four pre-meal and bedtime readings per day, CBG could have been obtained per primary team's discretion or nurse-initiated hypoglycemia protocol which recommends repeated CBG checking every 15 minutes until CBG recovers to >70 mg/dL. CBG readings were categorized by hospital meal distribution timing: pre-breakfast CBG from 5:00 am to 9:30 am, pre-lunch from 9:31 am to 4:00 pm, pre-dinner from 4:01 pm to 8:00 pm, and bedtime from 8:01 pm to 4:59 am. We defined hypoglycemia as CBG <70 mg/dL. CBG <40 mg/dL and >400 mg/dL were documented in our glucose database as <40 mg/dL and >400 mg/dL respectively and were represented as 39 mg/dL and 401 mg/dL respectively in our analysis.

**Insulin.** Insulin doses were represented as units/kg. Daily glargine and lispro doses were totaled per patient hospitalization day. Lispro doses were provided by breakfast, lunch, and dinner. Information on whether the lispro dose given was standing or supplemental was unavailable. Our inpatient hyperglycemia insulin protocol in the non-ICU setting is total daily dose per weight (kg x 0.5 units/kg) divided into 50% basal insulin/50% bolus insulin. Bolus is divided by 3 mealtimes. Supplemental insulin is added to bolus based on pre-meal glucose algorithm.

## Statistical analysis

We generated descriptive statistics for baseline demographic and clinical characteristics for each patient per hospitalization with and without GC use. GC use and dose were expressed per hospitalization day. For outcome measures, mean CBG and median insulin doses per hospitalization day were calculated. To compare baseline clinical characteristics and outcome measures between patients with and without GC, we used independent-samples t-test for normal continuous variables, Mann Whitney U test for non-normally distributed continuous variables, and $\chi2$ test (and Fisher's exact test if available) for categorical data. For hospitalization day measures, one-way repeated-measures ANOVA was used to analyze the differences between mean mealtime CBGs within each cohort; Friedman and Wilcoxon signed-rank tests were used to compare overall and individual median lispro doses within each cohort. Pearson's correlation was used to examine the association between insulin and GC dose, and the association between CBG and GC dose. Analysis was performed using SPSS Version 24 (SPSS Inc., Chicago, IL, USA). All tests of significance were two tailed, with $\alpha$-levels of 0.05.

## Results

### Eligible hospitalizations cohorts

Our initial criteria for extraction identified 8431 hospitalizations. After eliminating additional hospitalizations not meeting our inclusion criteria, 137 patients were in the GC group and 906 patients in the non-GC group (Fig 1). After matching for age (±10 years), gender, race, and HbA1c (±0.5%) [26], there were 131 patients for both GC and non-GC groups (Fig 1). Including up to day 7 of the hospitalizations, the GC group had data points for 419 hospitalization days (394 excluding days without GC treatment), and the non-GC group had 646 hospitalization days.

### Baseline characteristics

Cohort characteristics are shown in Table 1. There were no significant differences in age, gender, ethnicity, race, BMI, HbA1c, or length of stay between the two cohorts with 54.2% being African American and 60.3% being female.

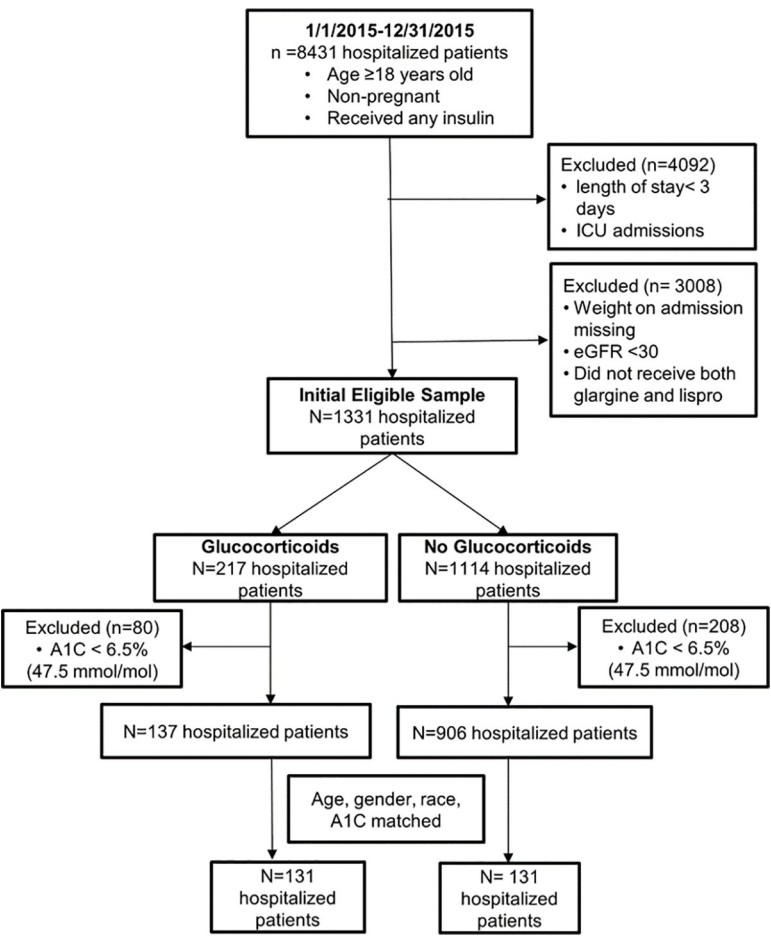

**Fig 1. CONSORT diagram.**

### Glucocorticoid types, doses, and administration times

In patients with GC, the median total prednisone equivalent dose received per day was 53.36 mg (IQR 30.00–80.04) or 0.59 mg/kg (IQR 0.35–1.03). GC used by type were 51.8% prednisone, 21.5% dexamethasone, 19.8% methylprednisolone, and 6.3% hydrocortisone. Timing of GC administration was: 55.4% between 6:00 am to 11:59 am, 22.6% between 12:00 pm to 5:59 pm, 12.5% between 6:00 pm to 11:59 pm, and 9.6% between 12:00 am to 5:59 am. The median number of days for GC per hospitalization were 3 days (IQR 1–4).

### Insulin doses

The median total daily glargine dose was similar in the GC vs non-GC group (0.24 units/kg (IQR 0.14–0.37) vs. 0.24 units/kg (0.17–0.34), P = 0.94). The median total daily lispro dose was 20% higher in the GC group (0.24 units/kg (IQR 0.13–0.30) vs. 0.20 units/kg (0.14–0.37), P = 0.06), albeit non-significantly. The median total daily insulin dose (glargine + lispro) was 18% non-significantly higher in the GC group (0.53 units/kg (IQR 0.32–0.79) vs. 0.45 units/kg (0.31–0.67), P = 0.21).

As shown in Fig 2, the median lispro doses at breakfast and lunch were similar in the GC vs. non-GC groups, breakfast: 0.10 units/kg (IQR 0.06–0.15) vs. 0.10 units/kg (0.06–0.13), P = 0.37; lunch: 0.10 units/kg (IQR 0.07–0.15) vs. 0.10 units/kg (0.06–0.14), P = 0.53. However,

**Table 1. Demographic and clinical characteristics after matching for race, gender, age, and HbA1c.**

| | GC (N = 131) | No GC (N = 131) |
|---|---|---|
| **Gender** | | |
| Female | 60.3% | 60.3% |
| Male | 39.7% | 39.7% |
| **Age** (mean years) ± SD | 58.5 ± 10.9 | 59.4 ± 10.7 |
| **Race** | | |
| African American | 54.2% | 54.2% |
| White | 32.8% | 32.8% |
| Others | 13.0% | 13.0% |
| **Ethnicity** | | |
| Hispanic/Latino/Spanish Origin | 29.0% | 29.0% |
| Non-Hispanic/Latino/Spanish Origin | 71.0% | 71.0% |
| **Weight** (mean kg) ± SD | 90.1 ± 29.3 | 92.6 ± 26.2 |
| **BMI** (mean kg/m$^2$) ± SD | 32.8 ± 10.0 | 34.0 ± 9.1 |
| **HbA1c** | | |
| % | 8.5 ± 1.8 | 8.6 ± 1.8 |
| mmol/mol | 69.7 ± 19.9 | 70.5 ± 19.3 |
| **Length of Stay** (mean days) ± SD | 4.9 ± 2.9 | 5.5 ± 3.4 |

No significant differences for any variables between groups.

BMI = body mass index, GC = glucocorticoid

dinner lispro dose was 20% significantly higher in the GC than non-GC group (0.12 units/kg (IQR 0.08–0.17) vs. 0.10 units/kg (0.06–0.14), P <0.01).

In the GC group, the differences between mealtime lispro doses was significant overall (P = 0.01). Specifically, lispro doses at dinner (0.12 units/kg (IQR 0.08–0.17) were greater than breakfast (0.10 units/kg (0.06–0.15) and greater than lunch (0.10 units/kg (0.07–0.15), both P< 0.01, with no significant difference between breakfast and lunch. In the non-GC group, the differences between mealtime lispro doses were similar (P = 0.06).

## Capillary blood glucose pattern

Mean average daily CBG was significantly higher in the GC group (221±58 vs. 177±43 mg/dL, P<0.001). As shown in Fig 2, mean CBG at mealtimes and bedtime were significantly greater in the GC group, especially at dinner and bedtime: breakfast (190±70 vs. 161±46 mg/dL, P<0.001), lunch (199±77 vs. 180±54 mg/dL, P = 0.03), dinner (254±69 vs. 184±63 mg/dL, P<0.001), and bedtime (260±72 vs. 182±55 mg/dL, P<0.001).

In the GC group, the CBG readings between mealtimes and bedtime were different overall (P<0.001), mainly due to dinner CBG being 30% greater than at lunch (254±69 vs. 199±77 mg/dL, P<0.001). CBG differences between breakfast and lunch (190±70 vs. 199±77 mg/dL, P = 0.20) and between dinner and bedtime (254±69 vs. 260±72 mg/dL, P = 0. 22), however, were not significant.

In the non-GC group, the CBG readings between meal times and bedtime were also different overall (P<0.001) because CBG at lunch was greater than at breakfast (180±54 vs. 161±46 mg/dL, P<0.001). CBG differences between lunch and dinner (180±54 vs. 184±63 mg/dL, P = 0.54) and between dinner and bedtime were not significant (184±63 vs. 182±55 mg/dL, P = 1.00).

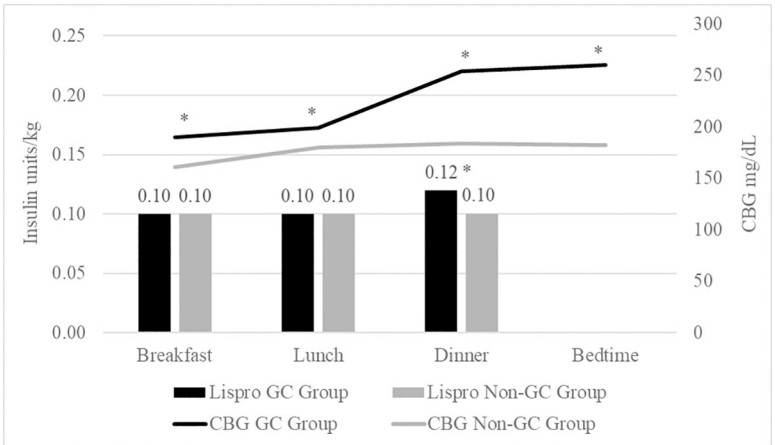

**Fig 2. Mean capillary blood glucose (CBG) readings at mealtimes and bedtime and median insulin lispro doses at mealtimes.** * P<0.05 between glucocorticoid (GC) and non-GC mealtime glucoses at same mealtime. ** P<0.05 between GC and non-GC lispro dose at dinner.

The occurrence of hypoglycemic episodes per hospitalization was lower in the GC than the non-GC group, significant at dinner (0.9 vs. 11.9%, P<0.001) and bedtime (0.0 vs. 5.9%, P = 0.03) but non-significant at breakfast (7.8 vs. 14.4%, P = 0.15) and lunch (8.3 vs. 14.4%, P = 0.16).

### Impact of GC dose on CBG and insulin dosing

GC dose was positively correlated with average CBG per day (R = 0.446, P<0.001). GC dose was also positively correlated total daily lispro dose/kg (R = 0.17, P<0.001) while GC dose did not correlate with total daily glargine dose/kg (R = 0.03, P = 0.58).

### Discussion

The study's intent was to evaluate the CBG pattern and insulin usage in hospitalized patients with DM with and without GC therapy. Patients with GC use exhibited significantly higher CBG at all mealtimes and bedtime, with the biggest increases being 38% higher at dinner and 43% higher at bedtime compared to non-GC group. Despite these high CBG readings, the GC group only received 20% more lispro at dinner than at lunch, which was not sufficient to control the bedtime CBG. This finding demonstrates that there is a need to increase further the prandial insulin doses beyond the 20% increase in this study.

As for the basal insulin, our data show that glargine dose/day was not different between groups treated with and without GC. Even though the fasting CBG in the GC group was significantly higher, this was likely a carryover from the higher bedtime CBG since our protocol does not recommend supplemental insulin at bedtime. Despite the higher pre-breakfast fasting CBG in the GC group, the glargine dose is still considered appropriate since the post bedtime CBG steeply declined overnight. Increasing the daily glargine dose thus may increase the risk of hypoglycemia.

Our findings of higher prandial insulin requirements with high GC use were consistent with the study led by Spanakis et al. 2014 [12], who demonstrated that double amounts of prandial insulin but similar basal glargine were required to achieve normoglycemia in 20 out of 58 hospitalized patients with DM receiving GC. However, no proportion of prandial insulin was delineated per meal. More specifically, Burt et al. 2015 [27] demonstrated that greater

delivery of insulin during the afternoon and evening was needed during methylprednisolone treatment in 24 hospitalized patients with type 2 DM; despite having received higher dosages of ultra-short-acting insulin between 1200 and 1700 hours (~10% more total per day), significantly higher CBG between 1700 and 2100 hours were observed.

Limitations of our study include the retrospective nature where potential unmeasured confounders could be present despite our best attempts in matching. Not included in our study database were the types or duration of DM, severity of underlying disease(s), in-patient nutritional status (on a diet or NPO), all of which could influence insulin requirements [12]. Because of the characteristics of our database, we were not able to separate the proportion of the prandial insulin and supplemental insulin. Finally, we were unable to give exact insulin dosing recommendations for attaining euglycemia since most hospitalizations with GC use did not achieve inpatient target glycemic control of 140–180 mg/dL [23].

Our study's strengths include well-matched cohorts, larger sample size than previous studies, insulin as the sole hyperglycemia medication, and availability of lispro doses at different meals with corresponding CBG data. This is the first study that shows that the major hyperglycemic effects of GCs begin at lunch and peaks at dinner.

Our study allowed the suggestion of an initial inpatient insulin regimen for patients with DM with GC use that is based on increasing the prandial insulin during lunch and dinner, reflecting the hyperglycemic GC pattern identified. The current protocol is 0.5 units/kg with 50% of the total dose being basal and 50% being prandial with the prandial dose divided equally for three meals. The protocol in this setting will need increased prandial doses for lunch and dinner. The particulars for protocol changes will need full corroboration in a prospective study that will test not only the achievement of targets but also their safety. An example of this could be the currently ongoing clinical trial by Cunningham et al. [28]. With the current Covid-19 pandemic increasing the use of inpatient steroids, the relevance of these adjustments take a larger footprint in our current view of inpatient hyperglycemia management.

## Supporting information

**S1 File. Cook county health inpatient hyperglycemia protocol.**
(PDF)

## Author Contributions

**Conceptualization:** Yu-Chien Cheng, Yannis Guerra, Michael Morkos, Chioma Onyenwenyi, Leon Fogelfeld.

**Data curation:** Yu-Chien Cheng, Yannis Guerra, Michael Morkos, Bettina Tahsin, Chioma Onyenwenyi, Louis Fogg, Leon Fogelfeld.

**Formal analysis:** Yu-Chien Cheng, Yannis Guerra, Bettina Tahsin, Louis Fogg, Leon Fogelfeld.

**Investigation:** Yu-Chien Cheng, Yannis Guerra, Michael Morkos, Chioma Onyenwenyi, Leon Fogelfeld.

**Methodology:** Yu-Chien Cheng, Yannis Guerra, Michael Morkos, Bettina Tahsin, Chioma Onyenwenyi, Louis Fogg, Leon Fogelfeld.

**Project administration:** Yu-Chien Cheng, Yannis Guerra.

**Supervision:** Yannis Guerra, Leon Fogelfeld.

**Validation:** Yu-Chien Cheng, Yannis Guerra, Michael Morkos, Bettina Tahsin, Leon Fogelfeld.

**Visualization:** Yu-Chien Cheng, Yannis Guerra, Michael Morkos, Bettina Tahsin, Leon Fogelfeld.

**Writing – original draft:** Yu-Chien Cheng, Yannis Guerra, Leon Fogelfeld.

**Writing – review & editing:** Yu-Chien Cheng, Michael Morkos, Bettina Tahsin, Chioma Onyenwenyi, Louis Fogg, Leon Fogelfeld.

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
