## [Decision Letter · Decision Letter 0]

7 Apr 2021

PONE-D-21-07298

Insulin Management in Hospitalized Patients with Diabetes Mellitus on High-Dose Glucocorticoids: Management of Steroid-Exacerbated Hyperglycemia

PLOS ONE

Dear Dr. Guerra,

Thank you for submitting your manuscript to PLOS ONE. After careful consideration, we feel that it has merit but does not fully meet PLOS ONE’s publication criteria as it currently stands. Therefore, we invite you to submit a revised version of the manuscript that addresses the points raised during the review process.

We look forward to receiving your revised manuscript.

Kind regards,

Sun Young Lee, MD

Academic Editor

PLOS ONE

Journal Requirements:

Reviewers' comments:

Reviewer's Responses to Questions

**Comments to the Author**

1. Is the manuscript technically sound, and do the data support the conclusions?

Reviewer #1: Partly

Reviewer #2: Partly

2. Has the statistical analysis been performed appropriately and rigorously? 

Reviewer #1: Yes

Reviewer #2: Yes

3. Have the authors made all data underlying the findings in their manuscript fully available?

Reviewer #1: Yes

Reviewer #2: Yes

4. Is the manuscript presented in an intelligible fashion and written in standard English?

Reviewer #1: Yes

Reviewer #2: Yes

5. Review Comments to the Author

Reviewer #1: In this paper, the authors performed a retrospective cohort study of hospitalized patients with diabetes to examine the effects of glucocorticoid use on blood glucose readings and insulin doses.

Strengths of this study include the standardized protocol where all patients are placed on a balanced basal bolus regimen and use of matching to control for age, gender, race, and hemogloblin A1c.

Limitations include its retrospective nature and reporting of data across all hospitalization days.

Specific comments:

How are insulin doses adjusted per protocol based on the blood sugar readings? I understand from the discussion that “the current protocol is 0.5 units/kg with 50% of total dose being basal (glargine) and 50% being prandial (lispro).” Is this applied to every patient on admission who has a GFR > 30, regardless of other factors (such as hemoglobin A1c, liver disease, glucocorticoid use, etc)?

From my understanding of this protocol, the starting dose of lispro would be 0.083 units/kg TID and 0.25 units/kg of glargine. The doses on hospital day 1 would be identical between the glucocorticoid exposed and unexposed groups and I suspect that differences in dosing between the groups would only start to emerge over time as the glucocorticoid exposed group’s insulin doses were increased in response to the blood sugars.

On page 7 it is reported “supplemental insulin is added to bolus based on pre-meal glucose algorithm.” Please include this algorithm in a supplemental figure. Furthermore, in the discussion there is mention that “our protocol does not recommend supplemental insulin at bedtime.” How is the glargine dose titrated per protocol? It is surprising that there was no association between the glucocorticoid dose and the glargine dose.

In the results, the median total daily glargine and lispro doses are compared between the exposed and unexposed groups. Does this include all of doses across every day of hospitalization? I would expect that the differences in the insulin doses to be larger after a few days of hospitalization where there has been titration since it sounds like both groups are started on the same weight-based dose on admission.

The authors report the median total daily lispro dose was 20% higher in the glucocorticoid exposed group compared with the unexposed group, which I suspect represents a comparison of the doses including all hospital days. It would be more meaningful to compare insulin doses after a few days of insulin titration, or on the last day of glucocorticoid use, or on the day of discharge, as the differences in insulin doses between the exposed and unexposed groups would likely be larger than on admission. The changes in insulin doses and blood sugars over time could be shown with a table of the median doses of glargine and lispro per day and median blood sugars vs. day of hospitalization stratified by exposure.

Was there any use of NPH or insulins other than lispro and glargine in this population? Were patients receiving continuous tube feeds excluded?

Reviewer #2: Comments to the editor:

Thank you for asking me to review this retrospective cohort study by Cheng and colleagues evaluating insulin doses and blood glucose values in non-critically ill, hospitalized patients with diabetes treated with glucocorticoids as compared to those not treated with glucocorticoids. This study adds to the literature regarding management strategies for steroid induced hyperglycemia in the inpatient setting, an area that has been relatively understudied and for which there is not yet an established best practice with respect to insulin dosing, particularly for rapid acting insulin. Prior to publication of their manuscript, several questions should be addressed.

1) Within the introduction, I would update the ADA Standards of Care reference to the 2021 version (of note there are no substantive changes in the recommendations but it is the most recent).

2) Within the methods section it is stated that the insulin protocol at the authors’ institution is dosed at 0.5u/kg/day and divided 50/50 basal and bolus for patients with GFR >30 ml/min and that those patients not dosed per protocol were excluded. The consort diagram suggests it is only patients who did not receive both basal and bolus insulin who were excluded and does not mention weight-based dosing. Can the authors please clarify if patients who received more or less than 0.5u/kg/day were excluded? I would assume given the insulin dosing results that there are patients included in the cohort who receive more or less insulin per this prescribed protocol.

3) The authors’ decided to convert glucocorticoids to prednisone equivalent doses and evaluate the impact on CBG and insulin dosing overall. Can they provide their rationale for this decision? It may be interesting to evaluate insulin dosing needs by type of glucocorticoid (specifically for prednisone and dexamethasone as these were the 2 most commonly used GC) as it has previously been demonstrated that prednisone and dexamethasone result in varying degrees of insulin resistance (Yasuda K et al, doi:10.1210/jcem-55-5-910).

4) Within the results section on insulin dosing, while glargine dosing was similar between groups it would be helpful for the authors to clarify which group corresponded to which doses.

5) Would specify that changes in insulin dosing and hypoglycemia were not statistically significant. I would argue that nearly 20% increase in insulin dose and nearly a doubling of hypoglycemia at breakfast are clinically significant findings. Would likely see statistical significance with a slightly larger sample size.

6) The last line of the limitations paragraph regarding requirement of further increase in bolus insulin feels out of place, would suggest that this be moved to earlier in the discussion (for example the first paragraph of discussion).

7) While the authors demonstrate higher rapid acting insulin doses at dinner and increased CBG at lunch and dinner in patients on GC, I do not believe their findings support the conclusion that a 30% increase in rapid acting insulin at lunch and 20% increase in rapid acting insulin at dinner may be recommended as this was not specifically evaluated in this study. Instead, I would argue that their data do suggest higher doses of bolus insulin are required in the mid-day and evening but that specific recommendations regarding percentage increase should be evaluated in a future prospective study. While they include a statement regarding this need for further evaluation of the proposed strategy, it should also be mentioned in the abstract conclusions and first paragraph of the discussion.

6. PLOS authors have the option to publish the peer review history of their article (what does this mean?). If published, this will include your full peer review and any attached files.

Reviewer #1: **Yes: **Dylan Thomas

Reviewer #2: No

---

## [Author Response · Author response to Decision Letter 0]

26 May 2021

Reviewers' comments:

Reviewer's Responses to Questions

 Comments to the Author

1. Is the manuscript technically sound, and do the data support the conclusions?

Reviewer #1: Partly

Reviewer #2: Partly

2. Has the statistical analysis been performed appropriately and rigorously?

Reviewer #1: Yes

Reviewer #2: Yes

3. Have the authors made all data underlying the findings in their manuscript fully available?

Reviewer #1: Yes

Reviewer #2: Yes

 

4. Is the manuscript presented in an intelligible fashion and written in standard English?

Reviewer #1: Yes

Reviewer #2: Yes

5. Review Comments to the Author

Reviewer #1: In this paper, the authors performed a retrospective cohort study of hospitalized patients with diabetes to examine the effects of glucocorticoid use on blood glucose readings and insulin doses.

Strengths of this study include the standardized protocol where all patients are placed on a balanced basal bolus regimen and use of matching to control for age, gender, race, and hemogloblin A1c.

Limitations include its retrospective nature and reporting of data across all hospitalization days.

Specific comments:

How are insulin doses adjusted per protocol based on the blood sugar readings? I understand from the discussion that “the current protocol is 0.5 units/kg with 50% of total dose being basal (glargine) and 50% being prandial (lispro).” Is this applied to every patient on admission who has a GFR > 30, regardless of other factors (such as hemoglobin A1c, liver disease, glucocorticoid use, etc)?From my understanding of this protocol, the starting dose of lispro would be 0.083 units/kg TID and 0.25 units/kg of glargine. The doses on hospital day 1 would be identical between the glucocorticoid exposed and unexposed groups and I suspect that differences in dosing between the groups would only start to emerge over time as the glucocorticoid exposed group’s insulin doses were increased in response to the blood sugars.

On page 7 it is reported “supplemental insulin is added to bolus based on pre-meal glucose algorithm.” Please include this algorithm in a supplemental figure. Furthermore, in the discussion there is mention that “our protocol does not recommend supplemental insulin at bedtime.” How is the glargine dose titrated per protocol? It is surprising that there was no association between the glucocorticoid dose and the glargine dose.

We do follow the in-patient protocol which has been added as supplemental material. Our protocol does use the liver disease status to adjust the insulin dose but not the A1c or the glucocorticoid. As mentioned in the Methods, those with GFR <30 were excluded with no change in protocol for those with GFR >30.

For the supplemental insulin and the glargine dose adjustment, please see supplemental material. The finding that the glargine was not associated with glucocorticoid dose seemed to be in line with previous evidence that steroids affect mostly post-prandial glucose (Eur J Endocrinol 2010;162:729-735)

In the results, the median total daily glargine and lispro doses are compared between the exposed and unexposed groups. Does this include all of doses across every day of hospitalization? I would expect that the differences in the insulin doses to be larger after a few days of hospitalization where there has been titration since it sounds like both groups are started on the same weight-based dose on admission.

As per our inpatient protocol, there were no initial differences in dosing between the groups. Analysis was done by dose per day with every day a unit of analysis as we expected the dosing to progressively increase daily as mentioned by the reviewer. 

The authors report the median total daily lispro dose was 20% higher in the glucocorticoid exposed group compared with the unexposed group, which I suspect represents a comparison of the doses including all hospital days. It would be more meaningful to compare insulin doses after a few days of insulin titration, or on the last day of glucocorticoid use, or on the day of discharge, as the differences in insulin doses between the exposed and unexposed groups would likely be larger than on admission. The changes in insulin doses and blood sugars over time could be shown with a table of the median doses of glargine and lispro per day and median blood sugars vs. day of hospitalization stratified by exposure.

We did not analyze glucose median during the entire hospitalization as we expected that those on steroids may have a longer length of stay resulting in a skewed analysis if compared day-by-day.

Was there any use of NPH or insulins other than lispro and glargine in this population? Were patients receiving continuous tube feeds excluded?

Per our inpatient protocol, NPH is primarily used on OB/GYN patients who were excluded from our study population. Insulin Regular is only used on IV drips in MICU. Continuous tube feeds in our hospital are primarily used in MICU patients who were excluded from the study.

Reviewer #2: Comments to the editor:

Thank you for asking me to review this retrospective cohort study by Cheng and colleagues evaluating insulin doses and blood glucose values in non-critically ill, hospitalized patients with diabetes treated with glucocorticoids as compared to those not treated with glucocorticoids. This study adds to the literature regarding management strategies for steroid induced hyperglycemia in the inpatient setting, an area that has been relatively understudied and for which there is not yet an established best practice with respect to insulin dosing, particularly for rapid acting insulin. Prior to publication of their manuscript, several questions should be addressed.

1) Within the introduction, I would update the ADA Standards of Care reference to the 2021 version (of note there are no substantive changes in the recommendations but it is the most recent).

The text and reference were updated.

2) Within the methods section it is stated that the insulin protocol at the authors’ institution is dosed at 0.5u/kg/day and divided 50/50 basal and bolus for patients with GFR >30 ml/min and that those patients not dosed per protocol were excluded. The consort diagram suggests it is only patients who did not receive both basal and bolus insulin who were excluded and does not mention weight-based dosing. Can the authors please clarify if patients who received more or less than 0.5u/kg/day were excluded? I would assume given the insulin dosing results that there are patients included in the cohort who receive more or less insulin per this prescribed protocol.

The assumption of the reviewer was correct. Our inpatient protocol allows to use a converted home dose of insulin. See supplemental material.

3) The authors’ decided to convert glucocorticoids to prednisone equivalent doses and evaluate the impact on CBG and insulin dosing overall. Can they provide their rationale for this decision? It may be interesting to evaluate insulin dosing needs by type of glucocorticoid (specifically for prednisone and dexamethasone as these were the 2 most commonly used GC) as it has previously been demonstrated that prednisone and dexamethasone result in varying degrees of insulin resistance (Yasuda K et al, doi:10.1210/jcem-55-5-910).

The rationale for converting steroids to prednisone equivalents was two-fold. First, it would permit us to compare patients across different therapies. Second, there are multiple patients who during the same hospitalization may have received different types of steroids. Due to our unit of analysis being day of hospitalization, the conversion allowed us to compare the different doses of different steroids on various days.

4) Within the results section on insulin dosing, while glargine dosing was similar between groups it would be helpful for the authors to clarify which group corresponded to which doses.

We clarified the groups in the beginning of Insulin Dosing in the Results section.

5) Would specify that changes in insulin dosing and hypoglycemia were not statistically significant. I would argue that nearly 20% increase in insulin dose and nearly a doubling of hypoglycemia at breakfast are clinically significant findings. Would likely see statistical significance with a slightly larger sample size.

We agree that the difference found is likely clinically significant yet within our available sample size, the current findings are as reported.

 

6) The last line of the limitations paragraph regarding requirement of further increase in bolus insulin feels out of place, would suggest that this be moved to earlier in the discussion (for example the first paragraph of discussion).

As suggested, the sentence has been moved to the first paragraph of Discussion.

7) While the authors demonstrate higher rapid acting insulin doses at dinner and increased CBG at lunch and dinner in patients on GC, I do not believe their findings support the conclusion that a 30% increase in rapid acting insulin at lunch and 20% increase in rapid acting insulin at dinner may be recommended as this was not specifically evaluated in this study. Instead, I would argue that their data do suggest higher doses of bolus insulin are required in the mid-day and evening but that specific recommendations regarding percentage increase should be evaluated in a future prospective study. While they include a statement regarding this need for further evaluation of the proposed strategy, it should also be mentioned in the abstract conclusions and first paragraph of the discussion.

We agree with the reviewers that the specific recommendations need to be studied and this was not the purpose of the study. We revised the manuscript accordingly in abstract conclusion and in the Discussion. 

6. PLOS authors have the option to publish the peer review history of their article (what does this mean?). If published, this will include your full peer review and any attached files.

Do you want your identity to be public for this peer review? For information about this choice, including consent withdrawal, please see our Privacy Policy.

Reviewer #1: Yes: Dylan Thomas

Reviewer #2: No

---

## [Decision Letter · Decision Letter 1]

18 Jun 2021

PONE-D-21-07298R1

Insulin management in hospitalized patients with diabetes mellitus on high-dose glucocorticoids: Management of steroid-exacerbated hyperglycemia

PLOS ONE

Dear Dr. Guerra,

Thank you for submitting your manuscript to PLOS ONE. After careful consideration, we feel that it has merit but does not fully meet PLOS ONE’s publication criteria as it currently stands. Therefore, we invite you to submit a revised version of the manuscript that addresses the points raised during the review process.

Please review the comments by both reviewers regarding the revision and include additional discussions as suggested by the reviewers.

We look forward to receiving your revised manuscript.

Kind regards,

Sun Young Lee, MD

Academic Editor

PLOS ONE

Journal Requirements:

Reviewers' comments:

Reviewer's Responses to Questions

**Comments to the Author**

1. If the authors have adequately addressed your comments raised in a previous round of review and you feel that this manuscript is now acceptable for publication, you may indicate that here to bypass the “Comments to the Author” section, enter your conflict of interest statement in the “Confidential to Editor” section, and submit your "Accept" recommendation.

Reviewer #1: All comments have been addressed

Reviewer #2: (No Response)

2. Is the manuscript technically sound, and do the data support the conclusions?

Reviewer #1: Yes

Reviewer #2: Partly

3. Has the statistical analysis been performed appropriately and rigorously? 

Reviewer #1: I Don't Know

Reviewer #2: Yes

4. Have the authors made all data underlying the findings in their manuscript fully available?

Reviewer #1: Yes

Reviewer #2: Yes

5. Is the manuscript presented in an intelligible fashion and written in standard English?

Reviewer #1: Yes

Reviewer #2: Yes

6. Review Comments to the Author

Reviewer #1: The authors have addressed my comments.

I would consider adding a reference to an ongoing randomized clinical trial to address The Management of Glucocorticoid-Induced Hyperglycemia in Hospitalized Patients: https://clinicaltrials.gov/ct2/show/NCT01810952

I would also add that there is some data to suggest that the insulin dosing pattern should be tailored to match the glucocortoid dosing and type (e.g. optimal dosing for prednisone may be different than for dexamethasone). Hence, a single protocol for all inpatients on steroids may not be optimal.

https://eje.bioscientifica.com/view/journals/eje/179/4/EJE-18-0315.xml

Reviewer #2: The authors have not yet sufficiently addressed the comment regarding the decision to convert glucocorticoid doses to prednisone equivalent within the text of the manuscript. The rationale provided in the response to reviewers should be included within the methods section.

7. PLOS authors have the option to publish the peer review history of their article (what does this mean?). If published, this will include your full peer review and any attached files.

Reviewer #1: **Yes: **Dylan Thomas

Reviewer #2: No

---

## [Author Response · Author response to Decision Letter 1]

28 Jul 2021

Reviewer #1: The authors have addressed my comments.

I would consider adding a reference to an ongoing randomized clinical trial to address The Management of Glucocorticoid-Induced Hyperglycemia in Hospitalized Patients: https://clinicaltrials.gov/ct2/show/NCT01810952

This ongoing trial is now referenced in the Discussion section. 

I would also add that there is some data to suggest that the insulin dosing pattern should be tailored to match the glucocortoid dosing and type (e.g. optimal dosing for prednisone may be different than for dexamethasone). Hence, a single protocol for all inpatients on steroids may not be optimal.

https://eje.bioscientifica.com/view/journals/eje/179/4/EJE-18-0315.xml

We’ve added reference to this approach in our Introduction.

Reviewer #2: The authors have not yet sufficiently addressed the comment regarding the decision to convert glucocorticoid doses to prednisone equivalent within the text of the manuscript. The rationale provided in the response to reviewers should be included within the methods section.

The rationale provided in our prior response is now included in the Methods section.

---

## [Decision Letter · Decision Letter 2]

13 Aug 2021

Insulin management in hospitalized patients with diabetes mellitus on high-dose glucocorticoids: Management of steroid-exacerbated hyperglycemia

PONE-D-21-07298R2

Dear Dr. Guerra,

We’re pleased to inform you that your manuscript has been judged scientifically suitable for publication and will be formally accepted for publication once it meets all outstanding technical requirements.

Kind regards,

Sun Young Lee, MD

Academic Editor

PLOS ONE

Additional Editor Comments (optional):

Reviewers' comments:

Reviewer's Responses to Questions

**Comments to the Author**

1. If the authors have adequately addressed your comments raised in a previous round of review and you feel that this manuscript is now acceptable for publication, you may indicate that here to bypass the “Comments to the Author” section, enter your conflict of interest statement in the “Confidential to Editor” section, and submit your "Accept" recommendation.

Reviewer #1: All comments have been addressed

Reviewer #2: All comments have been addressed

2. Is the manuscript technically sound, and do the data support the conclusions?

Reviewer #1: Yes

Reviewer #2: Yes

3. Has the statistical analysis been performed appropriately and rigorously? 

Reviewer #1: Yes

Reviewer #2: Yes

4. Have the authors made all data underlying the findings in their manuscript fully available?

Reviewer #1: Yes

Reviewer #2: Yes

5. Is the manuscript presented in an intelligible fashion and written in standard English?

Reviewer #1: Yes

Reviewer #2: Yes

6. Review Comments to the Author

Reviewer #1: All of my comments have been addressed.

All of my comments have been addressed.

All of my comments have been addressed.

Reviewer #2: The authors have addressed my comments in a satisfactory fashion. My recommendation is to accept the manuscript.

7. PLOS authors have the option to publish the peer review history of their article (what does this mean?). If published, this will include your full peer review and any attached files.

Reviewer #1: **Yes: **Dylan D. Thomas

Reviewer #2: No

---

## [Editor Report · Acceptance letter]

3 Sep 2021

PONE-D-21-07298R2 

Insulin management in hospitalized patients with diabetes mellitus on high-dose glucocorticoids: Management of steroid-exacerbated hyperglycemia 

Dear Dr. Guerra:

I'm pleased to inform you that your manuscript has been deemed suitable for publication in PLOS ONE. Congratulations! Your manuscript is now with our production department. 

Kind regards, 

on behalf of

Dr. Sun Young Lee 

Academic Editor

PLOS ONE